# Performance Assessment of GPM IMERG Products at Different Time Resolutions, Climatic Areas and Topographic Conditions in Catalonia

**Eric Peinó [1], Joan Bech [1,2,\*] and Mireia Udina [1]**

[1]  Department of Applied Physics—Meteorology, Universitat de Barcelona, 08028 Barcelona, Spain
[2]  Water Research Institute, Universitat de Barcelona, 08028 Barcelona, Spain
\*  Correspondence: joan.bech@ub.edu

**Abstract:** Quantitative Precipitation Estimates (QPEs) from the Integrated Multisatellite Retrievals for GPM (IMERG) provide crucial information about the spatio-temporal distribution of precipitation in semiarid regions with complex orography, such as Catalonia (NE Spain). The network of automatic weather stations of the Meteorological Service of Catalonia is used to assess the performance of three IMERG products (Early, Late and Final) at different time scales, ranging from yearly to sub-daily periods. The analysis at a half-hourly scale also considered three different orographic features (valley, flat and ridgetop), diverse climatic conditions (BSk, Csa, Cf and Df) and five categories related to rainfall intensity (light, moderate, intense, very intense and torrential). While IMERG_E and IMERG_L overestimate precipitation, IMERG_F reduces the error at all temporal scales. However, the calibration to which a Final run is subjected causes underestimation regardless in some areas, such as the Pyrenees mountains. The proportion of false alarms is a problem for IMERG, especially during the summer, mainly associated with the detection of false precipitation in the form of light rainfall. At sub-daily scales, IMERG showed high bias and very low correlation values, indicating the remaining challenge for satellite sensors to estimate precipitation at high temporal resolution. This behaviour was more evident in flat areas and cold semi-arid climates, wherein overestimates of more than 30% were found. In contrast, rainfall classified as very heavy and torrential showed significant underestimates, higher than 80%, reflecting the inability of IMERG to detect extreme sub-daily precipitation events.

**Keywords:** GPM-IMERG; satellite precipitation estimates; remote sensing; assessment; complex orography; extreme precipitation

## 1. Introduction

The effects of climate change on future precipitation remain uncertain [1]. However, climate model predictions simulate yearly decreases in semi-arid regions of the Mediterranean [2]. Mountain areas are particularly vulnerable, where the cryosphere is directly affected by global warming, which consequently leads to altered seasonal runoff patterns [3]. Thus, hydrological cycles will gradually shift from being dominated by snow and ice to being determined by rainfall [4]. Accurate precipitation measurements at different spatial and temporal scales are of great significance for validating numerical weather and climate models, managing water resources and predicting natural disasters.

However, quantitative estimates of precipitation often have significant uncertainty [5]. Rain gauges, which are the world's most common method of obtaining accurate and reliable measurements at high temporal resolutions, provide point-scale measurements. This makes them unable to fully capture the spatial variability of the precipitation or to capture extreme local events in many areas wherein the instrument density is low. Ground-based radar-derived estimates are another feasible method, but due to their poor

global coverage, the effects of terrain blockage [6] and the difficulties associated with estimating mixed and solid phase precipitation [5,7], there are limitations to obtain reliable estimates. Thus, satellite precipitation estimates (SPEs) offer an excellent way to compensate for some of these limitations and, although they have their own shortcomings, can be considered a complement to other methods [8].

Based on the success of the previous Tropical Rainfall Measurement Mission (TRMM), the Global Precipitation Measurement Mission (GPM) core satellite plus a constellation of satellites from partner countries provide one of the most accurate and fine-grained spatio-temporal resources for global precipitation measurements [9]. GPM has advanced sensors such as the GPM dual frequency precipitation radar (DPR) and microwave imager (GMI), which quantify precipitation more accurately and detect light and solid precipitation [10]. The associated processing, Integrated Multisatellite Retrievals for GPM (IMERG), incorporates, fuses and intercalibrates several infrared, microwave (MW) and gauge observations to provide precipitation estimates at relatively high spatial (0.1° × 0.1°) and temporal (30 min) resolutions [9].

Since the launch of the GPM (February 2014), the chronology of publications evaluating the performance of the IMERG reflects a growing trend of research interest in the subject [11]. Most of the works that take a country or region of study stratify the results of the validation process according to different time scales [12–18], topographic features [15,19–24], climatic conditions [23,25–27] and in terms of precipitation intensity [19,20,28–33]. In this way, a more specific description of IMERG behaviour under different conditions is obtained, leading to the choice of a more suitable use for its application.

Several investigations with this approach have been developed in Mediterranean countries. In Greece, Kazamias et al. [34] explored the performance of IMERG Final across the country at daily, seasonal and annual scales, in different elevation zones and rainfall intensities. Caracciolo et al. [35] studied the influence of morphology and land–sea transition on the reliability of IMERG Final at hourly and daily scales, while Chiaravalloti et al. [36] evaluated and compared the IMERG Early, Late and Final products over complex terrain in southeastern Italy. Tapiador et al. [37] introduced for the first time the results of a validation in Spain based on a comparison with a high-resolution grid of daily precipitation derived from the records of approximately 2300 rain gauges covering the Iberian Peninsula and the Balearic Islands. The study at annual, seasonal and daily resolutions also analysed the spatial structure of precipitation and considered different precipitation thresholds for the three IMERG products. Similarly, Navarro et al. [38] validated the IMERG at the south of the Pyrenees and the Ebro valley according to four parameters: altitude, climate type, seasonality and quality of surface observations. Finally, Tapiador et al. [39] selected the IMERG Late product to evaluate the consistency of ground observations and satellite data during the Storm 'Filomena' in January 2021. Pradhan et al. [11] recently reviewed validation studies of IMERG and identified the most common limitations in this type of work, offering some suggestions to solve them. An important pending issue is the evaluation of IMERG products at multiple time scales, including sub-daily periods, to understand the errors associated with temporal aggregation. Further analysis in mountainous regions, over different climatic regimes, geographical conditions and assessing the effect of rainfall intensity on their accuracy still require the attention of the scientific community.

Based on these research gaps, the objectives of this work are twofold: (1) To evaluate the precipitation estimates obtained from the three IMERG runs (Early, Late and Final) at different time scales (half-hourly, hourly, daily, monthly, seasonal and annual) simultaneously taking as reference the automatic stations of the Meteorological Service of Catalonia and (2) To validate the IMERG estimates at the highest temporal resolution (30 min) according to different orographic features (valley, flat, ridgetop), different climatic conditions (BSk, Csa, Cf, Df) (see Appendix B) and according to different precipitation intensity thresholds (light, moderate, heavy, very heavy, torrential). The study considers the period from 2015 to 2020, so that full calendar years of the GPM core satellite data are

employed. We focus on the region of Catalonia, northeast of the Iberian Peninsula, being one of the first studies to evaluate the behaviour of IMERG at a sub-daily temporal resolution in this region. This complements the previous studies done in IP and other areas with complex orography.

Sections 2.1 and 2.2 provide a description of the study area and details of the methodology, data and assessment metrics employed. Sections 3.1 and 3.2 compare the rain gauge observations and estimates of the three IMERG products at different time scales simultaneously. Sections 3.4 and 3.5 focus on the semi-hourly scale, considering different orographic and climatic conditions as well as different precipitation intensity thresholds, respectively. The most significant results are discussed in Section 4, and finally a summary with the most relevant aspects is given in Section 5.

## 2. Materials and Methods

### 2.1. Study Area

Catalonia is a region wherein topographic complexity and high climatic variability are a challenge for the remote sensing estimation of precipitation from satellite- or ground-based products, as well as for the estimation of the precipitation field using rain gauge stations [38,40]. The area of study is in the north-east (NE) of the Iberian Peninsula with approximately 32,107 km² and over 580 km of coastline facing northeast to southwest towards the Mediterranean Sea (Figure 1a). It is bordered to the north by the Pyrenees (Figure 1a), a mountainous barrier that connects the Iberian Peninsula with the European mainland and has elevations that can exceed 3000 masl. Another distinctive feature is the Central Depression (Figure 1a), characterized by flat land with few orographic contrasts resulting from the erosion of the Ebro and its tributaries.

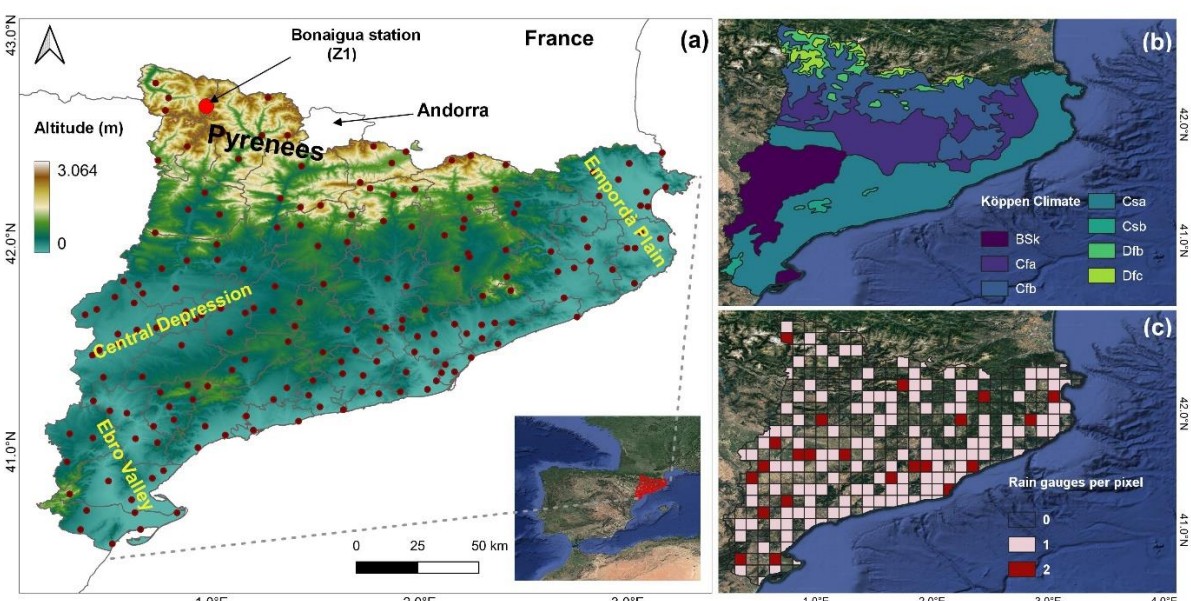

**Figure 1.** (**a**) Digital elevation model of Catalonia and XEMA stations network distribution. (**b**) Köppen climate classification in the study area. (**c**) Number of XEMA stations per IMERG pixel in the Catalonia domain.

The location of the orographic features and the pronounced topographic gradient of the region influence atmospheric low-level circulations and, particularly, the rainfall distribution over the entire territory [41,42]. On a large scale, it is an area of contact between air masses of different characteristics: cold or polar, coming from mid and high latitudes, and warm or tropical, typical of subtropical and tropical latitudes. The

northwestern side of the Pyrenees (Köppen types Dfb, Dfc), which is exposed to the influence of humid air masses from the Atlantic, is where the highest annual accumulations are observed, with average values exceeding 1200 mm. The coastal and pre-coastal mountain chains (Csa) enhance the pluviometric effects of the Mediterranean cyclogenesis along the coast and form a pluviometric screen on the rest of the territory [43].

In inland areas, the climatic regime is highly conditioned by the precipitation deficit, which barely exceeds 400 mm per year. In the Central Depression (BSk), the winter is relatively cold, with frequent fog favoured by thermal inversions [44], and the summer is hot and dry. This impact of the general circulation patterns of the atmosphere modulated by the complex topography of the region promotes heavy rainfall, frequent flash floods and complex mesoscale meteorological events [42,45,46].

### 2.2. Datasets

#### 2.2.1. IMERG V06B Data

This study validates data obtained from 2015 to 2020 obtained by the Integrated Multisatellite Retrievals for GPM (IMERG) version 06B at different time scales. GPM (2014–present) under the IMERG algorithm calibrates, fuses and interpolates precipitation estimates from various passive microwave sensors, infrared sensors and monthly rain gauge records [47] every 30 min, at a spatial resolution of 0.1° × 0.1° and with a global coverage from −90°S to 90°N latitude.

The IMERG system provides three products: Early (latency of ~4 h after observation and forward propagation only), Late (latency of ~14 h after observation and includes forward and backward propagation) and Final run (~3.5 months after observation, using both forward and backward propagation and including monthly gauge analysis). The Final run also uses a month-to-month adjustment, which combines the multisatellite data for the month with the Global Precipitation Climatology Centre (GPCC) gauge (1° × 1° grid), derived from approximately 6700 stations worldwide [38]. Its influence in each half-hour slot is a ratio multiplier that is fixed for the month, but spatially varying [9].

IMERG data were obtained in UTC time and were downloaded through the NASA Goddard Earth Sciences Data and Information Services Center (GES DISC) [48]. Precipitation estimation data (combined microwave–infrared in the Early and Late products and precipitation estimates with post-processing gauge calibration in the Final product ("*PrecipitationCal*" variable, in all cases)) were analysed. Initially, the data had a resolution of 30 min and was aggregated at different time intervals: hourly, daily, monthly, seasonal and annual.

#### 2.2.2. XEMA Data

The validation of the different IMERG products was conducted taking as a reference rainfall data from the automatic stations network (XEMA) managed by the Meteorological Service of Catalonia [49]. Semi-hourly rainfall records with a resolution of 0.1 mm were obtained in UTC time, between 1 January 2015 and 31 December 2020. Quality control applied to rain gauge data includes comparisons with close stations and correlation analysis [50,51]. From these initial data, hourly accumulation was generated, in which we verified that the data from the two 30-minute intervals corresponding to the hour did exist. Two criteria were applied to perform the comparison between the XEMA and IMERG data. The first criterion (Criterion 1) consists of requiring that there are at least 80% of records for each tested time scale. The second criterion (Criterion 2) restricts the comparison to couples of IMERG and XEMA data equal to or greater than 0.1 mm (this threshold is explained in Section 2.3.1). The results of applying these criteria are shown in Appendix A (Table A1). This distribution means that of the 417 IMERG pixels covering the region, 40% contain at least one rain gauge and 5% contain two rain gauges for validation (Figure 1c). The GPCC rain gauges used to calibrate the IMERG Final come

from first order stations of the AEMET network [38], so all our data are independent from those used for calibration.

### 2.3. Methodology

2.3.1. Overview

Figure 2 shows a diagram summarizing the validation process of the three IMERG products based on the comparison with ground-based observations from XEMA rain gauges. To overcome the spatial mismatch between the two datasets, a pixel-to-point method [20,28] was applied to obtain the satellite information at each coordinate of the meteorological stations. This method allowed for a direct pairwise comparison between the rainfall data and the IMERG pixel value where the station is located. In case there was over one rain gauge in an IMERG pixel, the independence of the precipitation records in each one was maintained for the comparison. This method offers us the advantage of avoiding additional uncertainties derived from interpolation, considering the complexity of the topography in the region. Finally, the information from 164 IMERG pixels was associated with the 186 rain gauges, which corresponds to an overall density of 1.13 rain gauges per 100 km$^2$. This value represents more than six times the threshold recommended by the World Meteorological Organisation (WMO) for the interior flat and undulating areas [52].

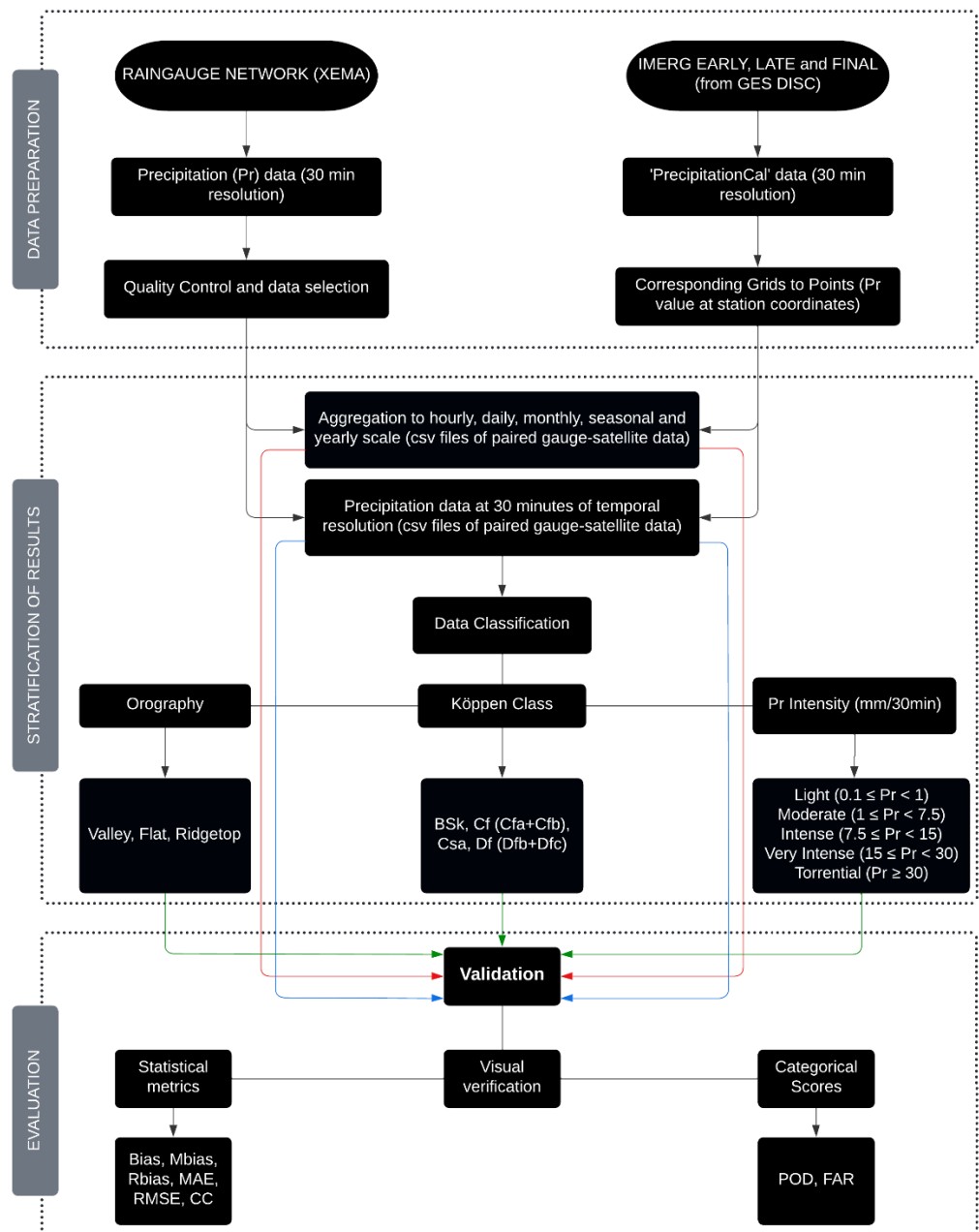

**Figure 2.** Schematic methodology applied in the data preparation, classification, and validation.

　　　The first part of the study focused on evaluating and comparing the performance in the three IMERG products (Early, Late and Final) at multiple time scales: half-hourly, hourly, daily, monthly, seasonal, annual and annual mean over the period of 2015–2020. The different datasets were obtained from the aggregation of the semi-hourly precipitation accumulations (mm), considering only those records with at least 0.1 mm in both the IMERG and XEMA products (Criterion 2 described in Section 2.2.2). Note that 0.1 mm is the minimum rainfall detected by XEMA rain gauge. In this way, only precipitation periods are considered, and no further biases are introduced due to the different minimum precipitation amounts provided by each dataset, as discussed by Trapero et al. [42] in their Appendix A.

　　　The second part of the research focuses only on the evaluation of the IMERG products on a half-hourly time scale and under various classifications. In order to achieve the

classifications, the IMERG pixels were grouped and classified according to common orographic features and Köppen climatic conditions (Table A2).

The stratification of the results according to orography was based on a 5 m DEM [53] of the region of Catalonia. For each pixel, the topographic position index (TPI) was calculated [54] and with the tool "Corridor Designer" [55,56] a raster file was obtained in which each grid was classified as valley (TPI ≤ −12 m), flat (−12 m < TPI < 12 m, slope < 6°) and ridgetop (TPI ≥ 12 m).

Similarly, the process to divide the domain according to different climatic conditions started from a vector file with the Köppen classification in Catalonia [57], which was rasterised at a high spatial resolution (0.01°) to better preserve the vector characteristics. Four climatic categories were thus determined: BSk, Cf (fusion of Cfa and Cfb), Csa and Df (fusion of Dfb and Dfc).

Finally, the raster files were resampled to IMERG resolution using the so-called majority interpolation method [58] and the corresponding labels were extracted at the station level at both resolutions (initial high resolution and IMERG resolution). The station points where the orographic and climatic labels at different spatial resolutions coincided were taken for the IMERG evaluation process. Figure 3 shows the distribution of the pixels and weather stations used for validation, according to the category they represent.

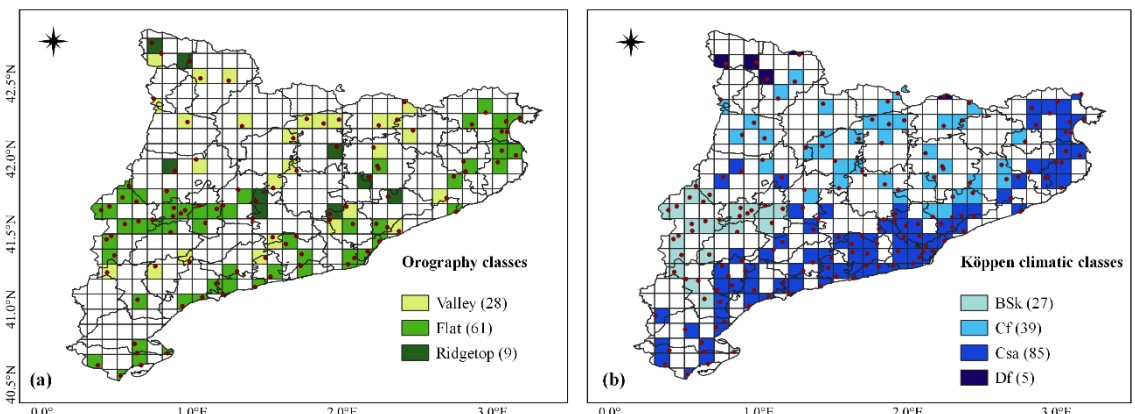

**Figure 3.** Distribution of IMERG pixels and stations (red dots) classified according to (**a**) orography (**b**) Köppen climate classification used for validation. The numbers in brackets represent the number of stations under that classification.

Half-hourly XEMA data were classified into five categories of precipitation intensity: light, moderate, heavy, very heavy and torrential (Figure 2). These categories were obtained by scaling the thresholds in mm/h established by AEMET [59].

2.3.2. Categorical and Continuous Verification Scores

To validate IMERG's ability to detect rainfall events correctly, categorical verification scores calculated from a 2 × 2 contingency table classifying events exceeding thresholds are used (Table 1). The recognition of the different possible situations (hits, false alarms, true positives, and misses) was done for various intensity thresholds. The categorical verification scores used were the probability of detection (*POD*) and the false alarm rate (*FAR*) (Table 2). The *POD* represents the proportion of events correctly detected by IMERG out of the total observed rainfall events, while the *FAR* represents the fraction of false detected rainfall events.

**Table 1.** Contingency table for comparing rainfall observed by XEMA and estimated by IMERG for a given threshold.

| Estimated Rainfall | Observed Rainfall | |
|---|---|---|
| | Gauge Rain ≥ Threshold | Gauge Rain < Threshold |

| IMERG rain ≥ threshold | *Hits (H)* | *False alarms (F)* |
|---|---|---|
| IMERG rain < threshold | *Misses (M)* | *Correct Negatives* |

**Table 2.** List of categorical verification metrics used to evaluate IMERG products.

| Name | Formula | Perfect Score |
|---|---|---|
| Probability of detection (*POD*) | $POD = \dfrac{Hits}{Hits\ +\ Misses}$ | 1 |
| False alarm ratio (*FAR*) | $FAR = \dfrac{False\ alarms}{False\ alarms\ +\ Hits}$ | 0 |

Additional continuous statistical metrics were used (Table 3). The Spearman correlation coefficient, used in cases such as this wherein there is no normality or homoscedasticity in the data, ranges from −1 to 1 and measures of the monotonicity of the relationship [60] between the IMERG and XEMA estimates. We also calculated the confidence interval for this statistic and tested for statistical significance at 95% of confidence. The other five metrics are used to quantify the associated error. *Bias* is a measure of the average error between IMERG and XEMA, while *Rbias* describes the systematic error. Positive (negative) values of *Bias* and *Rbias*, as well as those greater than unity (less than unity) of *Mbias*, denote the overestimation (underestimation) by the satellite products. The *MAE* shows the average magnitude of the absolute errors and, finally, the *RMSE* measures the magnitude of the average error, giving more weight to large errors without indicating the direction of deviation between IMERG and XEMA.

**Table 3.** List of the continuous verification metrics used to evaluate IMERG products.

| Name | Formula | Unit | Perfect Score |
|---|---|---|---|
| Spearman's correlation coefficient | $r = \dfrac{cov(R(S_i), R(O_i))}{\sigma_{R(S_i)}\, \sigma_{R(O_i)}}$ | - | 1 |
| Mean error (*Bias*) | $Bias = \dfrac{1}{n}\sum_{i=1}^{n}(S_i - O_i)$ | mm | 0 |
| Relative bias (*Rbias*) | $Rbias = \dfrac{\sum_{i=1}^{n}(S_i - O_i)}{\sum_{i=1}^{n} O_i} \times 100$ | % | 0 |
| Multiplicative bias (*Mbias*) | $Mbias = \dfrac{\sum_{i=1}^{n} S_i}{\sum_{i=1}^{n} O_i}$ | - | 1 |
| Mean absolute error (*MAE*) | $MAE = \dfrac{\sum_{i=1}^{n}|S_i - O_i|}{n}$ | mm | 0 |
| Root mean square error (*RMSE*) | $RMSE = \sqrt{\dfrac{1}{n}\sum_{i=1}^{n}(S_i - O_i)^2}$ | mm | 0 |

$S_i$ is the value of satellite/model precipitation estimates for the $i^{th}$ event, $O_i$ is the value of rain gauge observation for the $i^{th}$ event, $n$ is the number of observed records, $cov(R(S_i), R(O_i))$ is the covariance of the rank variables, $\sigma_{R(S_i)}$ and $\sigma_{R(O_i)}$ are the standard deviations of the rank variables.

## 3. Results

### 3.1. Mean Annual Precipitation 2015–2020

A comparison of mean annual precipitation amounts was made between IMERG products and XEMA data. Figure 4 shows the spatial distribution of the mean annual precipitation of IMERG products compared with rain gauge recorded from 2015 to 2020. In addition, the probability of occurrence of annual precipitation and the kernel density estimation (KDE) curve associated with the distribution of each dataset are plotted (lower panel).

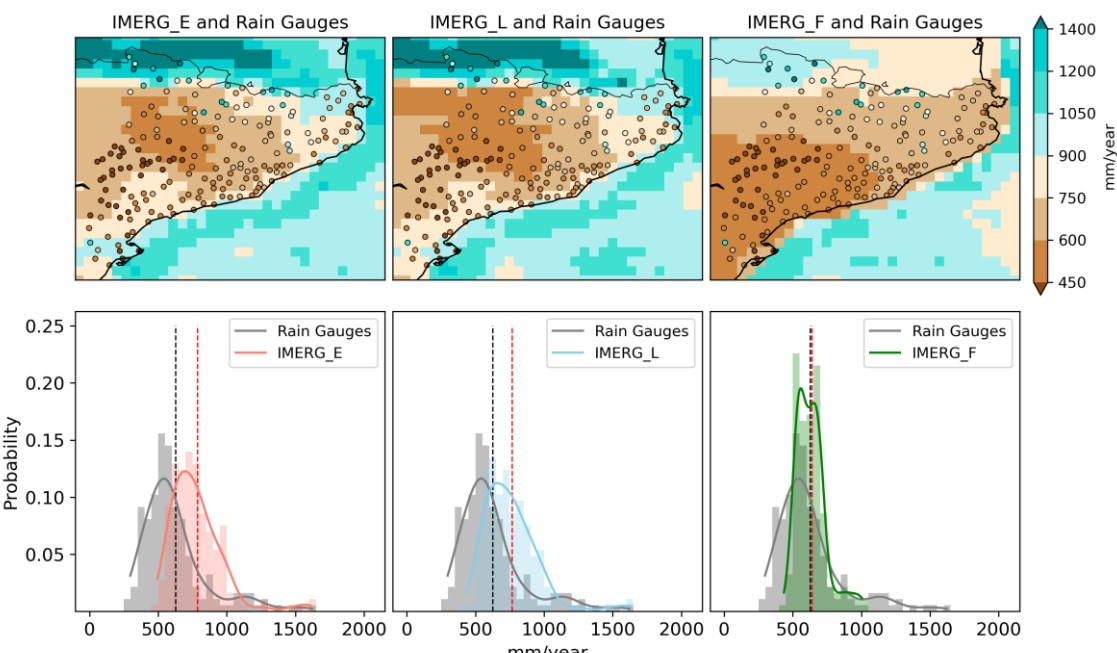

**Figure 4.** (**Top panel**) Mean annual precipitation accumulations of IMERG products and XEMA stations, in the period of 2015–2020. (**Bottom panel**) KDE curve associated with the distribution of each dataset, the black (red) dashed line represents the mean of the XEMA observations (IMERG).

According to rain gauge data, the average annual rainfall in Catalonia during this period varies between 300 mm and 1600 mm/year. The lowest records are observed in the Central Depression, where they do not exceed 450 mm/year, followed by the coastal areas with values around 600 mm/year. In contrast, stations close to the Pyrenees have accumulations usually exceeding 900 mm/year and those located above 2000 masl typically are over 1600 mm/year, which represents the maximum values for the region under study, and also some of the highest of the Iberian Peninsula. This high spatial variability is consistent with previous precipitation climatologies [61–63] in the studied region, which guarantees the representativeness of the selected sample.

The comparative analysis between the products shows a very similar performance between IMERG_E and IMERG_L, while in IMERG_F there is evidence of the unbiased effect thanks to the calibration with GPCC rainfall. It is also worth noting that the three IMERG products broadly reproduce the spatial rainfall pattern in the region, characterized by a marked latitudinal gradient that decreases from north to south. However, there are discrepancies in magnitude that are substantial. IMERG_E and IMERG_L overestimate precipitation by over 20% in almost all the territory with biases of 160 and 140 mm/year, respectively. This overestimation is notable in the areas of the Central Depression, characterized by a dry continental climate with low pluviometric values. Similar results were detected by Kazamias et al. [34], wherein the *IMERG_unCal* show the largest discrepancies in the areas of Greece with low annual accumulations. Similarly, Navarro et al. [38] also found a general overestimation of precipitation over the Ebro Delta river, and Tapiador et al. [37] reported an underestimation in the Pyrenees mountain massif.

Although the tendency of IMERG_E and IMERG_L to overestimate is shown in the same way at the pre-coastal, coastal and Ebro basin areas, the correction carried out in IMERG_F is effective and generally reflects annual mean values very similar to the rain gauge records (Figure 4, bottom panel). However, IMERG_F generally reduces and smooths the precipitation field over the Pyrenees and some high-altitude stations show an increased bias exceeding 600 mm/year.

### 3.2. Continuous Verification Scores for Different Time Scales

Table 4 shows a summary of various statistics calculated at half-hourly, hourly, daily, monthly and annual scales considering all valid records between 2015 and 2020. The *Bias*, *MAE* and *RMSE* are standardized to the mean of the observations at the different time scales, which allows for comparisons to be made between them.

**Table 4.** List of the continuous verification metrics used to evaluate IMERG products.

| | N | Bias (mm) | Mbias | Rbias (%) | MAE (mm) | MAE (%) | RMSE (mm) | RMSE (%) | CC |
|---|---|---|---|---|---|---|---|---|---|
| | | | | 30 min | | | | | |
| IMERG_F | 277616 | −0.07 | 0.95 | −4.85 | 1.19 | 87.36 | 2.37 | 173.30 | 0.33 |
| IMERG_L | 277616 | 0.20 | 1.15 | 14.59 | 1.39 | 101.76 | 2.70 | 197.15 | 0.29 |
| IMERG_E | 277616 | 0.26 | 1.19 | 18.86 | 1.49 | 109.18 | 2.89 | 211.11 | 0.23 |
| | | | | Hourly | | | | | |
| IMERG_F | 199255 | −0.05 | 0.98 | −2.16 | 1.88 | 87.27 | 3.51 | 162.81 | 0.37 |
| IMERG_L | 199255 | 0.39 | 1.18 | 18.25 | 2.23 | 103.26 | 4.21 | 195.35 | 0.33 |
| IMERG_E | 199255 | 0.42 | 1.20 | 19.60 | 2.35 | 109.01 | 4.46 | 206.85 | 0.26 |
| | | | | Daily | | | | | |
| IMERG_F | 70399 | −0.12 | 0.99 | −1.44 | 6.22 | 72.62 | 10.68 | 124.66 | 0.58 |
| IMERG_L | 70399 | 1.71 | 1.20 | 19.94 | 7.93 | 92.56 | 14.68 | 171.42 | 0.53 |
| IMERG_E | 70399 | 1.57 | 1.18 | 18.35 | 8.01 | 93.56 | 14.72 | 171.91 | 0.49 |
| | | | | Monthly | | | | | |
| IMERG_F | 12802 | 0.81 | 1.02 | 1.53 | 20.32 | 38.50 | 30.60 | 57.97 | 0.85 |
| IMERG_L | 12802 | 11.75 | 1.22 | 22.27 | 33.17 | 62.84 | 51.13 | 96.87 | 0.67 |
| IMERG_E | 12802 | 13.44 | 1.25 | 25.46 | 33.79 | 64.01 | 51.49 | 97.55 | 0.66 |
| | | | | Spring | | | | | |
| IMERG_F | 996 | −3.65 | 0.98 | −1.97 | 48.25 | 26.03 | 70.15 | 37.85 | 0.83 |
| IMERG_L | 996 | 8.02 | 1.04 | 4.33 | 75.46 | 40.71 | 101.30 | 54.65 | 0.54 |
| IMERG_E | 996 | 6.61 | 1.04 | 3.57 | 73.81 | 39.82 | 100.31 | 54.12 | 0.56 |
| | | | | Summer | | | | | |
| IMERG_F | 1020 | 11.39 | 1.10 | 9.64 | 43.41 | 36.74 | 59.73 | 50.55 | 0.85 |
| IMERG_L | 1020 | 97.23 | 1.82 | 82.28 | 105.47 | 89.26 | 143.32 | 121.29 | 0.65 |
| IMERG_E | 1020 | 97.84 | 1.83 | 82.80 | 106.46 | 90.10 | 142.63 | 120.70 | 0.62 |
| | | | | Autumn | | | | | |
| IMERG_F | 1032 | 2.34 | 1.01 | 1.15 | 52.09 | 25.55 | 70.80 | 34.73 | 0.80 |
| IMERG_L | 1032 | 33.69 | 1.17 | 16.53 | 84.27 | 41.33 | 109.55 | 53.73 | 0.61 |
| IMERG_E | 1032 | 46.89 | 1.23 | 23.00 | 89.53 | 43.91 | 114.42 | 56.12 | 0.61 |
| | | | | Winter | | | | | |
| IMERG_F | 820 | −2.42 | 0.98 | −1.91 | 37.79 | 29.83 | 60.58 | 47.82 | 0.91 |
| IMERG_L | 820 | 7.77 | 1.06 | 6.14 | 56.20 | 44.36 | 93.27 | 73.62 | 0.83 |
| IMERG_E | 820 | 14.11 | 1.11 | 11.14 | 54.51 | 43.03 | 88.75 | 70.06 | 0.84 |
| | | | | Yearly | | | | | |
| IMERG_F | 6204 | 9.65 | 1.02 | 1.55 | 139.36 | 22.35 | 194.17 | 31.14 | 0.86 |
| IMERG_L | 6204 | 139.76 | 1.22 | 22.41 | 226.11 | 36.26 | 280.06 | 44.92 | 0.60 |
| IMERG_E | 6204 | 159.22 | 1.26 | 25.54 | 230.12 | 36.91 | 285.82 | 45.84 | 0.63 |

In terms of *Rbias*, IMERG_E and IMERG_L present an overestimation of precipitation close to 20% at all time scales, except at the seasonal and yearly levels. In contrast, this behaviour only occurs in IMERG_F at monthly and annual scales, although it does not exceed 2%. At daily and sub-daily scales, IMERG_F slightly underestimates precipitation relative to observations, with values ranging between −0.05 mm/h and −0.12 mm/day,

which is relatively small compared to the mean of the observations at these scales (2.16 mm and 8.56 mm, respectively).

The analysis of the average error (*Bias*) reflects a significant improvement in the IMERG_F at all scales, although much more appreciable at the monthly and annual scales. At the latter, the *Bias* decreases to 9.65 mm compared to the 159.22 mm recorded by IMERG_E, which means a reduction of the error close to 90%. At the monthly scale, the error value decreases by about 16 times compared to the Early and Late products. While this significant error reduction could represent a good indicator of the improvement in precipitation estimation with IMERG_F, we must consider the limitations of this statistic and its relationship with the possible cancellation of positive and negative errors [64] between IMERG_F and ground-based observations.

As expected, as the temporal resolution decreases, there is a decrease in the *MAE* and the normalised *RMSE* regarding the mean for all products, with few differences at sub-daily scales. This behaviour is most evident in IMERG_F, in which the *MAE* decreases from 0.87 mm at 30 min to 0.22 mm at the annual scale, and the *RMSE* decreases from 1.73 mm to 0.31 mm. This improvement with a lower scale can be seen in the Taylor plot shown in Figure 5a, which displays the *STD*, *CC* and centred *RMSE* statistics normalised to the standard deviation of the three products for all temporal resolutions. A clear improvement in IMERG_F is observed at the monthly and annual scales, with values close to the benchmark (correlation and standard deviation equal to 1). The worst results are shown at the sub-daily scales with low correlation values and in the Early and Late products, with standard deviations higher than the benchmark unit. These differences between IMERG_F and the rest of the products, which grow with the increasing scale, highlight a gradual improvement as more information is integrated into the algorithm. Finally, while this product is expected to provide the most reliable estimates for research [47], the other two products can also be used for related to low latency applications [65].

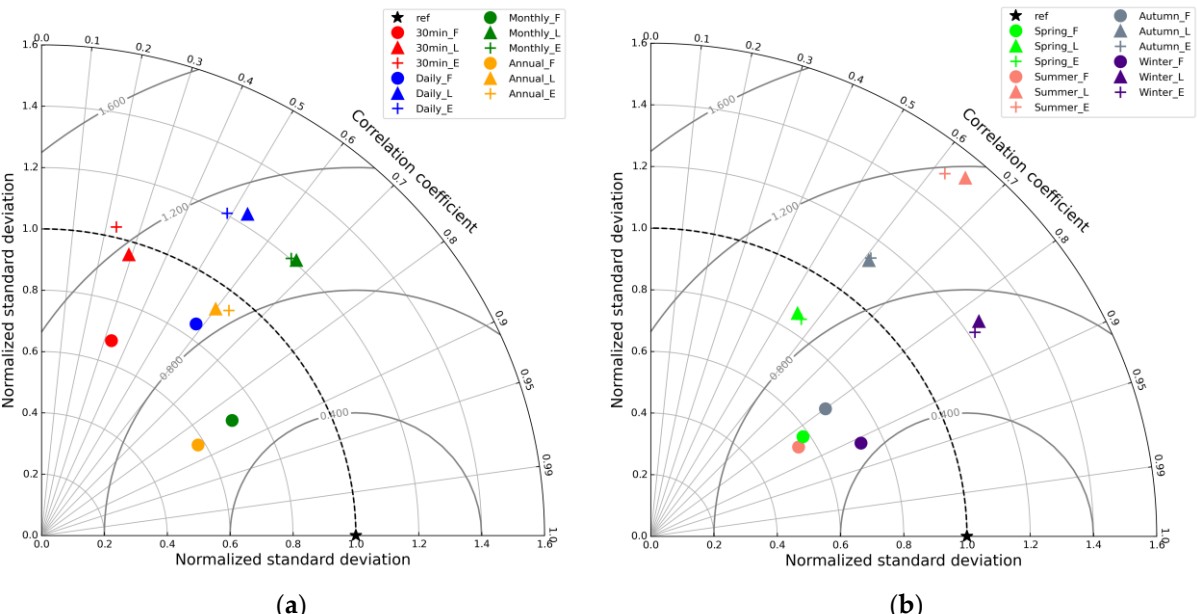

**Figure 5.** (**a**) Taylor diagram at sub-daily, daily, monthly, and annual scales of the products IMERG_E, IMERG_L and IMERG_F. (**b**) Same as the (**a**) figure but shows seasonal scales.

In the seasonal analysis (Figure 5b), IMERG_E and IMERG_L overestimate the precipitation values substantially. These errors are more noticeable during the summer period with a systematic error of over 97 mm and *MAE* and *RMSE* values around 105 mm and 143 mm, respectively. Interestingly, in this period, IMERG_F introduces significant improvements that reduce the overestimation to less than 10%, but it is still the season of

the year wherein the worst results are obtained. Precipitation in the summer months is low throughout the Iberian Peninsula and Catalonia, but local storms with convective development usually occur, wherein the amount of precipitation fallen is not adequately captured by IMERG.

The values of the errors in autumn, although lower than in summer, also show overestimates of precipitation in all products and *MAE* and *RMSE* values, which, even with the unbiasing of the Final product, remain relatively high (*MAE* equal to 52.09 mm and *RMSE* equal to 70.80 mm). A similar behaviour is observed in the rest of the seasons of the year, although the *RMSE* values practically double the *MAE* values, which may be caused by the occurrence of extreme phenomena and bring into play the sensitivity of this statistic in such records.

IMERG_F reproduces the annual cycle of precipitation relatively accurately, identifying the spring and autumn months as those that make the overall greatest contribution to the annual cycle amount, while the winter and summer months show the lowest accumulations with very few differences between them. On the other hand, IMERG_E and IMERG_L represent the summer period as the second highest contribution with an average of approximately 215 mm, higher than that recorded in the observations (118 mm), which is consistent with the overestimation made by these products during this period.

The correlation coefficient calculated at the different time scales showed statistical significance at 95% of confidence in all cases. In Figure 5a, at sub-daily scales, similar correlation values are shown among all products, and although a slight improvement appears in IMERG_F, it does not exceed 0.37. IMERG_E and IMERG_L at scales higher than daily show moderate linear correlations close to 0.6, and it is IMERG_F that represents high correlations, higher than 0.8. Similarly, there is a decrease in the standard deviation, closer to the reference point (*STD* = 1), as a result of the unbiasing to which it is subjected. The performance shown demonstrates that this product would be the most suitable for the analysis of precipitation at seasonal and annual scales.

### 3.3. Categorical Verification Scores for Different Time Scales

Figure 6 shows a summary of the contingency table verification score at the different time scales. For each dataset shown, a threshold greater than or equal to the mean of the observations recorded at each time step is applied.

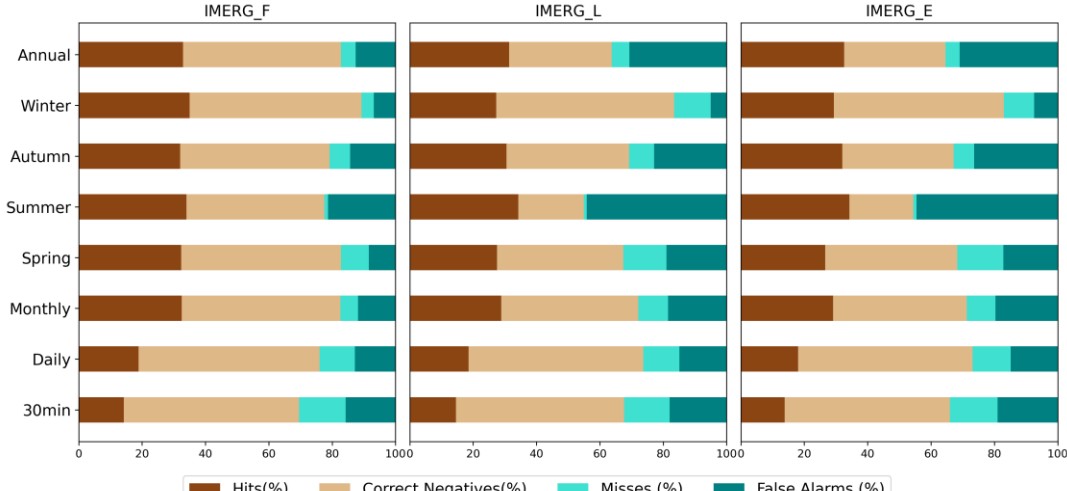

**Figure 6.** Fraction of events detected as hits, false alarms, misses and correct negatives for the three IMERG products at different time scales. The thresholds selected for each time scale coincide with the mean of the observations at that scale: Half-hourly (1.4 mm), Daily (8.6 mm), Monthly (52.8 mm), Spring (185.8 mm), Summer (118.8 mm), Autumn (203.9 mm), Winter (126.7 mm) and Annual (623.5 mm).

As shown in Figure 6, IMERG_F has a higher ability to detect correct negatives with values close to 50% at all scales, although IMERG_L and IMERG_E are also very similar at sub-daily and daily scales. The percentage of hits tends to increase at scales higher than daily, while the percentage of misses decreases. According to the selected thresholds, the ability of IMERG to estimate precipitation is affected by the detected false alarms. These represent the highest percentage during the summer period in IMERG_E and IMERG_L.

Figure 7 provides the performance of the *POD*, and the *FAR* values at different time scales for different precipitation thresholds. The error associated with the calculation of the statistic at each point, as outlined by Jolliffe and Stephenson [64], is also shown. The figure shows a clear improvement of IMERG estimates as time scale increases. At a half-hourly scale, the ability of IMERG to estimate events at different thresholds is remarkably poor.

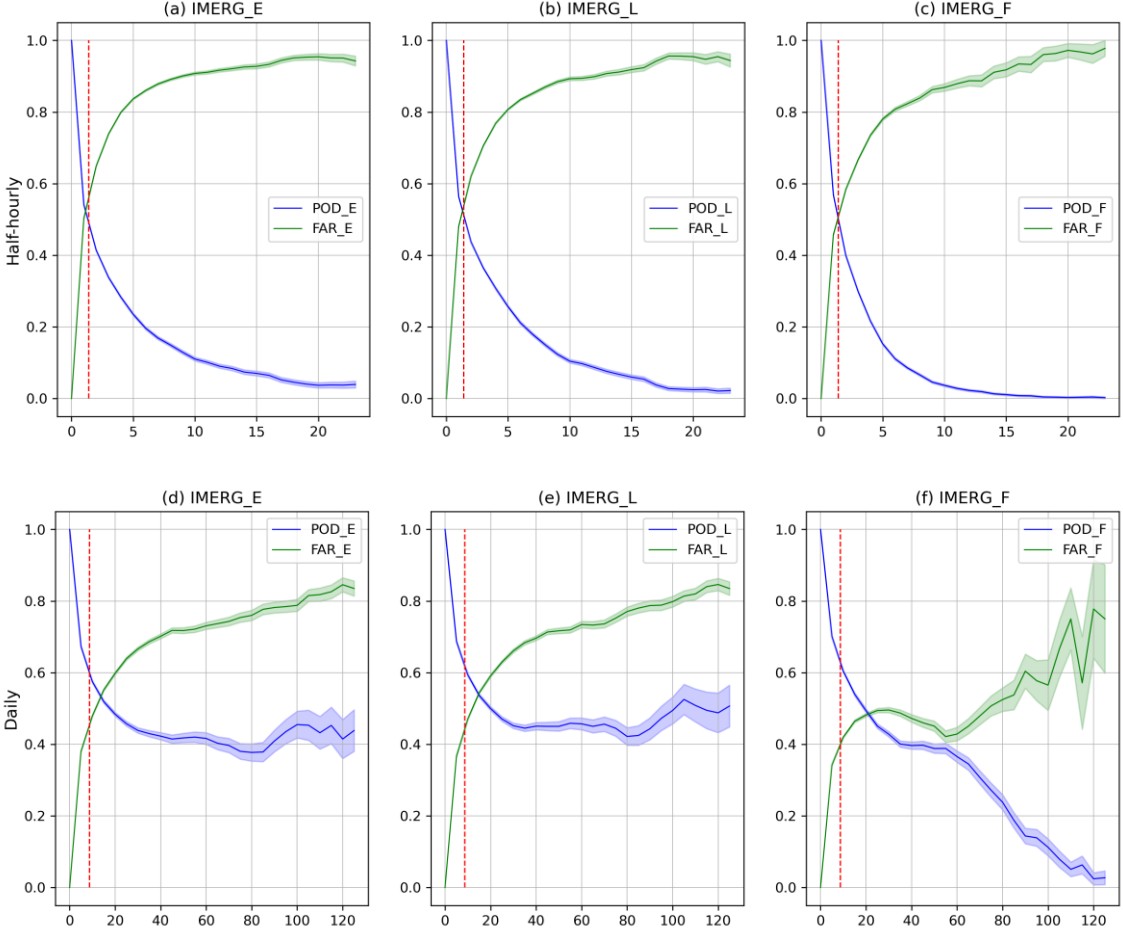

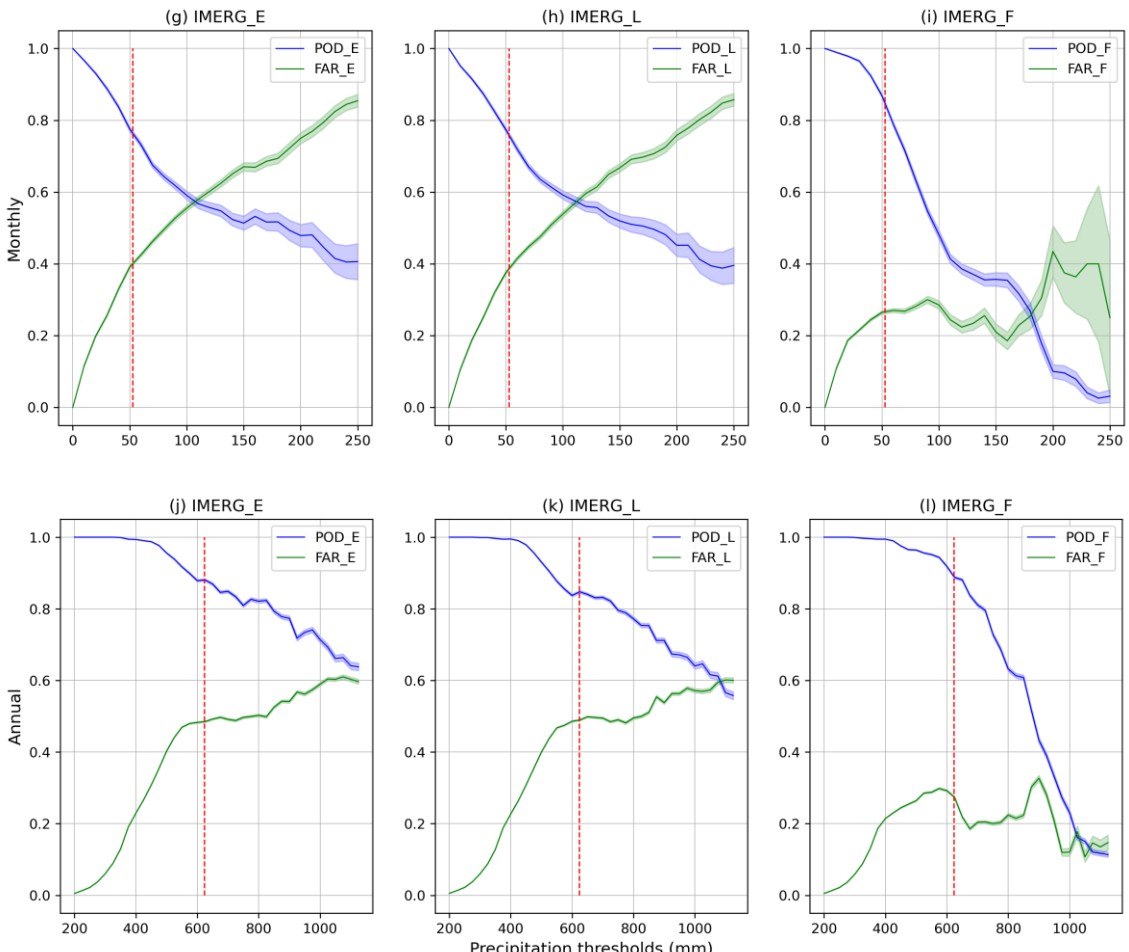

**Figure 7.** *POD*, *FAR* and errors associated at different time scales and precipitation thresholds. The vertical dashed red line represents the mean of the observations at each time scale.

If we consider the cut-off point between the *POD* and *FAR* line as a limit from which the satellite shows some decay in its event detection ability, we observe that it increases considerably with decreasing temporal resolution. At daily scales, this cut-off point occurs at around 20 mm/day, a value well above the 75th percentile of the sample, which indicates a much better performance compared to the estimation of events at the half-hourly scale, where a value slightly lower than the mean of the recorded observations at this scale is observed (1.4 mm/30 min).

Similarly, for the analysis at monthly and annual scales with cut-off points above 100 mm and 1000 mm, respectively, it ensures the correct identification of rainfall events up to this threshold. As higher thresholds of rainfall in the domain are assessed, the ability of the satellite decreases, highlighting the difficulty of IMERG to detect rainfall extremes at any scale.

Although the three products behave similarly, IMERG_F gives worse results, especially on scales above the monthly scale, wherein the *POD* values decrease faster than in the rest of the products, which is associated with the unbiasing effect induced by the calibration of GPCC stations. These results are consistent with Shawky et al. [66], which found no significant improvement of IMERG_F over IMERG_E in the arid environment of Oman. This result is in line with Sharifi et al. [16], Behrangi et al. [67] and Gosset et al. [68] when positing that the gauge adjustment product (IMERG_F) can change the precipitation amounts, but it cannot modify the occurrence of precipitation.

*3.4. Half-Hourly IMERG Products for Different Terrain and Climate Conditions*

This section will test the abilities and shortcomings of the three IMERG products at a high temporal resolution (30 min). In addition, differences in the estimation of precipitation by satellite products will be analysed when considering the terrain over which they are estimated and under different climatic conditions.

Figure 8 shows the differences over each station between IMERG and the rainfall records of the XEMA network. In valley areas, the analysis of the systematic error shows a marked underestimation of precipitation in IMERG_F, with mean values of −0.15 mm/30 min, which represents an underestimation of 10% regarding the rain gauges. IMERG_L and IMERG_E show a tendency to overestimate the accumulated values and show *MAE* and *RMSE* values even higher than 100% relative to the mean.

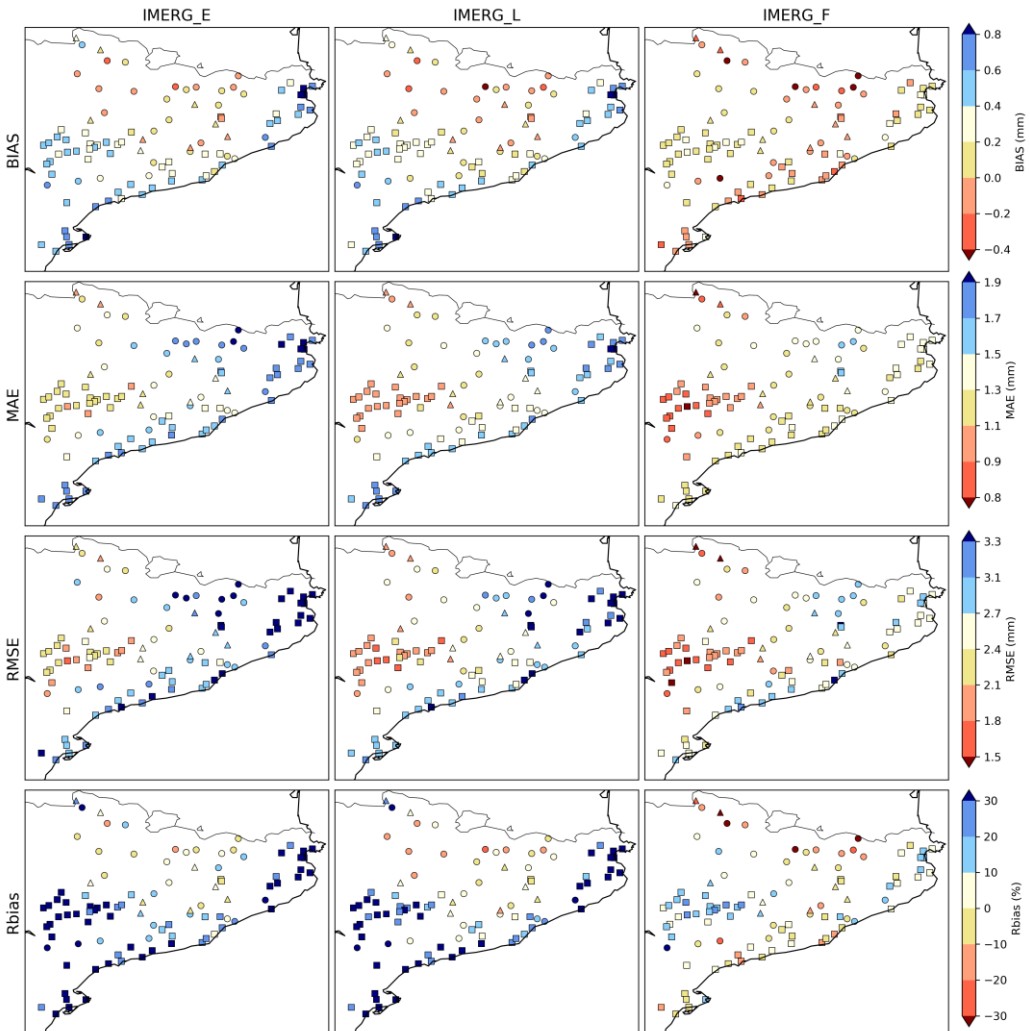

**Figure 8.** Distribution of Bias, MAE, RMS and Rbias errors at each station point and according to the orography type where they are located: Ridgetop (triangle), Flat (square), Valley (circle).

There is a more marked tendency in the behaviour of IMERG in areas representing ridgetops. While IMERG_E and IMERG_L overestimate precipitation, and this could be verified in all time scales, the effect of the calibration incorporated in IMERG_F causes a significant smoothing, such that the *Rbias* reaches critical values lower than −30% sometimes, as in the Bonaigua station (Z1) (Figure 2) at 2266 masl This marked underestimation and change in behaviour from one product to another is probably related to the low density of GPCC reference stations in high altitude areas for calibration. The *CC* shows a pattern in all three products with poor values, barely exceeding 0.3.

The largest errors occur in the stations in flat areas (Flat) with an average bias higher than 0.4 mm/30 min and *Rbias* values higher than 30% in IMERG_E and IMERG_L (Figure 8). Although IMERG_F significantly decreases the error, the tendency to overestimate the values is still maintained, and under this terrain classification the highest *MAE* and *RMSE* values regarding the mean are obtained (higher than 100% and 200%, respectively). In these areas, 43% of the automatic stations are located and analysed, which corresponds to the entire central inland part of the region of Catalonia, the coastal strip, the Ebro basin and the north-western part of the territory. This plays a significant role in the global results regardless of terrain classification.

Figure 9 evaluates the *Rbias* of the three IMERG products under different climatic classifications. For example, IMERG_E presents a large overestimation over the BSk stations and IMERG_F shows a high underestimation over the Df stations. Overall, a clear improvement in bias reduction is found for BSk and Csa stations when IMERG_F is compared to IMERG_E. The improvement is not so evident for the Cf stations, and in contrast, there is a clear bias increase for the Df stations. These results obtained for 30 min records coincide with previous studies by Navarro et al. [38] in the Ebro basin for seasonal and annual scales

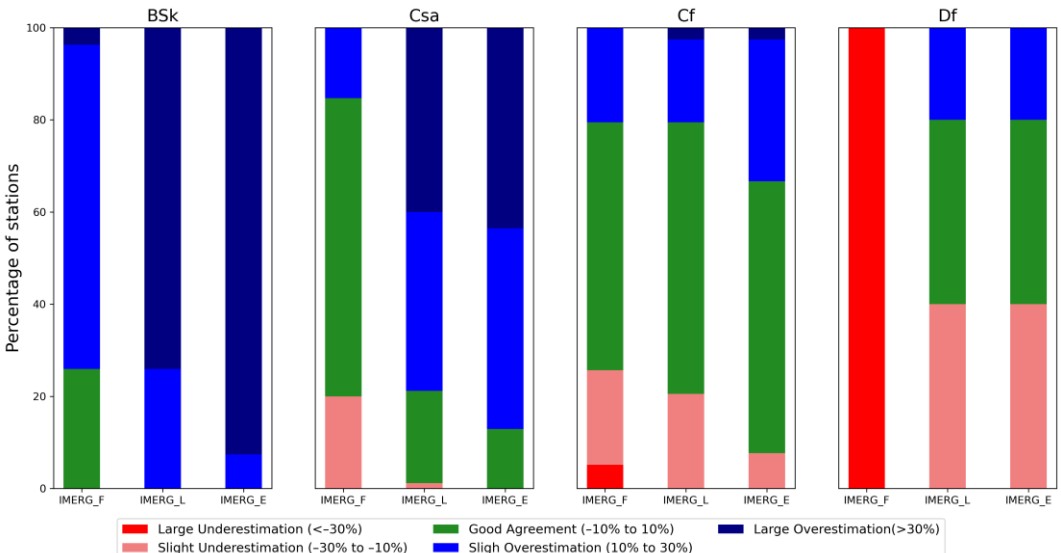

**Figure 9.** Stacked bars of the half-hourly relative error (*Rbias*) computed for each group of station for each climatic group. The colours represent the five categories of *Rbias* described in the legend.

*3.5. Intensity*

Table 5 shows a summary of the statistics obtained in the validation process of three IMERG products, considering five categories of rainfall intensity recorded in 30 min. The categories of light, moderate, intense, very intense and torrential rain were scaled from a previous classification of rainfall intensity in 1 h, according to sources from the Spanish Meteorological Agency (AEMET) [59].

**Table 5.** Summary of statistics calculated according to the intensity of rainfall recorded by rain gauges in 30 min.

| | N | BIAS (mm) | Mbias | Rbias (%) | MAE (mm) | RMSE (mm) |
|---|---|---|---|---|---|---|
| light (0.1 ≤ Pr < 1) | | | | | | |
| IMERG_F | 177039 | 0.56 | 2.35 | 134.83 | 0.70 | 1.25 |
| IMERG_L | 177039 | 0.76 | 2.81 | 181.30 | 0.90 | 1.81 |
| IMERG_E | 177039 | 0.85 | 3.04 | 203.89 | 1.00 | 2.06 |
| moderate (1 ≤ Pr < 7.5) | | | | | | |

| | | | | | | |
|---|---|---|---|---|---|---|
| IMERG_F | 94589 | −0.62 | 0.74 | −25.68 | 1.55 | 2.15 |
| IMERG_L | 94589 | −0.28 | 0.88 | −11.54 | 1.81 | 2.70 |
| IMERG_E | 94589 | −0.27 | 0.89 | −11.31 | 1.91 | 2.89 |
| heavy (7.5 ≤ Pr < 15) | | | | | | |
| IMERG_F | 4553 | −7.37 | 0.28 | −71.98 | 7.55 | 8.12 |
| IMERG_L | 4553 | −6.36 | 0.38 | −62.12 | 7.07 | 7.79 |
| IMERG_E | 4553 | −6.56 | 0.36 | −64.04 | 7.34 | 8.05 |
| very heavy (15 ≤ Pr < 30) | | | | | | |
| IMERG_F | 1296 | −16.54 | 0.16 | −83.65 | 16.63 | 17.32 |
| IMERG_L | 1296 | −14.89 | 0.25 | −75.32 | 15.18 | 16.16 |
| IMERG_E | 1296 | −15.07 | 0.24 | −76.23 | 15.41 | 16.40 |
| torrential (Pr ≥ 30) | | | | | | |
| IMERG_F | 139 | −32.57 | 0.11 | −89.47 | 32.57 | 33.19 |
| IMERG_L | 139 | −29.63 | 0.19 | −81.40 | 29.63 | 30.70 |
| IMERG_E | 139 | −28.98 | 0.20 | −79.60 | 28.98 | 30.53 |

The results obtained show substantial overestimation discrepancies for all rainfall intensity categories and in all IMERG products. Light rainfall, represented by the highest number of records, is overestimated by twice as much *Mbias* by IMERG_F and nearly three times as much by the rest of the products. This implies a relative error rate (*Rbias*) higher than 100% in all cases and a systematic error significantly higher than the mean of the observations. The best performance based on the *MAE* and *RMSE* is obtained by IMERG_F, although they are still quite high compared to the average of the studied records. Such indicators of overestimation in this category have been reported in previous studies [23,27].

On the contrary, at precipitation thresholds above 1 mm/30 min (moderate, heavy, very heavy and torrential), IMERG shows a tendency to underestimate precipitation, which becomes more significant as the intensity of precipitation increases (Figure 10). For the classes heavy, very heavy and torrential, the satellite shows errors ranging between −60% and −90% of the deficit in relation to the rain gauges. The systematic errors in these groups are similar in magnitude to the mean absolute errors and to the mean of the values recorded by the stations in each of the corresponding thresholds, which register a more realistic, significant underestimation.

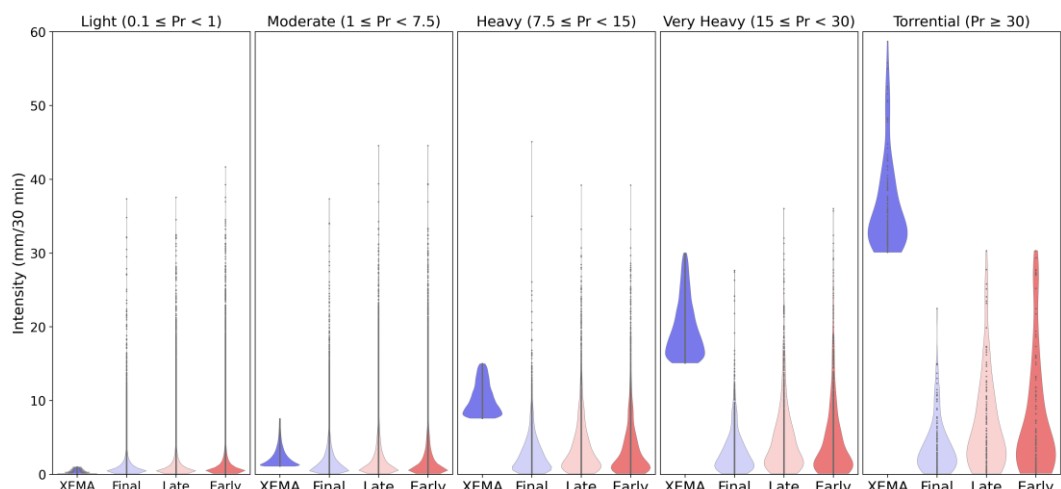

**Figure 10.** Violin plots of half-hourly rain gauge observations (XEMA) and IMERG products for the five rainfall intensity classes considered. Rainfall rate thresholds are given in mm/30 min.

Among the three products, IMERG_F provides the worst results, while IMERG_L presents the best values, although these differences are not marked. These results are in agreement with studies by Mazzoglio et al. [69] and show the challenge of detecting precipitation extremes at this resolution. Many of these extremes occur in the form of short and local intense rainfall, so they cannot be correctly captured due to the spatial and temporal resolution of satellite sensors. Precipitation at the daily and sub-daily scales is much more variable than monthly precipitation, and regional effects such as topography and local circulation play an important role in rainfall occurrence and distribution [16].

## 4. Discussion

In line with previous work, IMERG roughly reproduces the spatial pattern and temporal variability of rainfall in the region of study [17,24,70]. However, there are differences in the magnitude estimation for the different run types: Early, Late and Final. While there is a tendency to overestimate the accumulations in the Early and Late products across the whole territory, IMERG_F reduces the errors and shows a better ability to estimate the amount of precipitation at all time scales, with higher accuracy at monthly, seasonal and annual scales. However, for this product, there is again a tendency to underestimate in areas with complex topography, i.e., high mountain areas such as the Pyrenees. This result is reasonable and has been reported in other high mountain areas [17]. Navarro et al. [38] and Tapiador et al. [37] suggested that this may be due to the lack of rain gauges contributing to the GPCC in high altitude areas, as well as to the low resolution of the GPCC grid (1° × 1°), which makes detection difficult in areas wherein precipitation is highly variable at small scales. Finally, Navarro et al. [38] also mentioned the reduced detection capacity of IMERG in the identification of convective orographic rainfall, mainly related to mesoscale factors.

At the seasonal scale, a similar underestimation is observed in the Final product at all temporal scales. However, in the Early and Late products, significant errors appear during the summer with a tendency to overestimate the cumulates producing high *MAE* and *RMSE* errors. This differs from the studies of Moazam and Najafi, [13] and Navarro et al. [71], wherein the worst results were obtained mainly during winter, when the ground surface is covered with snow and ice [24]. However, our results are in line with Retalis et al. [15], in which the best results were obtained in the rainy seasons (winter and autumn). In semi-arid areas, the summer period is represented by low precipitation values, which makes detection by satellite sensors difficult [17]. Another important issue to consider is that precipitation can be affected by a high rate of evaporation, where some of the liquid water evaporates during the fall process and is no longer part of the effective precipitation [26,31,72], a virga being the extreme case wherein no precipitation reaches the ground. This phenomenon, coupled with the fact that satellite retrievals of precipitation are based on the structure of cloud systems [73] and may not adequately account for the level of evaporation, may lead to the overestimation of precipitation in arid regions.

The effect of not accounting for evaporation in semi-arid areas further explains that, in terms of precipitation event detection, the error in IMERG is dominated by the occurrence of false alarms, especially in summer. In the occurrence of typical deep convective clouds with relatively cold cloud tops (anvils) and, with the absence of PMW measurements, the IR algorithm may falsely assign precipitation to pixels with cold brightness temperature values [74]. Furthermore, in terms of IMERG's ability to detect events given a continuous threshold of cumulates, no significant improvement of one product over the other is observed. In fact, IMERG_E and IMERG_L offer better performance as the thresholds grow with more stable *POD* and *FAR* values and lower uncertainty in the statistics. This is related to the inability of IMERG_F to detect extremes, similarly associated to the calibration of GPCC.

Few studies include the validation of IMERG at the highest temporal resolution (30 min). Even so, the authors of [26,75] agree on the decreased estimation capability of the

three products with increasing temporal resolution. The repetition time of the GPM and the downscaling and interpolation procedures to 30 min [76] are some of the main causes of the errors obtained. At this scale, the largest errors occur in flat areas, which coincides with the BSk climate, with a tendency to overestimate. The authors of [38] found that in these areas, IMERG tended to overestimate precipitation equally. These regions, mainly represented in our study by inland depressions (Ebro valley) and coastal areas, are affected by extreme precipitation events occurring at local scales. Orographic factors and mesoscale conditions generate an uneven distribution of precipitation over the territory, resulting in a very spatially uncorrelated precipitation field [37] and therefore an added challenge for satellite estimates.

Finally, the overestimation of lightprecipitation associated with the detection of false alarms and the underestimation of precipitation extremes reflects a similar behaviour to that found in the Tibetan Plateau [27]. Along these lines, it is important to be aware of the limitations of the assessment procedure, which may influence the accuracy of the results. Firstly, it is worth mentioning that the rain gauge data used were not corrected for the effect of wind, so the measurements may suffer from systematic biases caused by wind-induced evaporation loss and the underestimation of trace values [24]. On the other hand, in terms of the pixel-to-point method, although it has advantages over other methods [70], it is very difficult for a (point-scale) rain gauge to represent the actual precipitation situation in an IMERG pixel-scale range. These inherent differences between the rain gauge estimate and the precipitation in the satellite area can directly influence the high values of false alarms, as well as the detection of extreme precipitation events occurring at the local scale. Especially in a region like Catalonia, characterised by its orographic complexity and climatic variability, more rain gauges per IMERG cell may provide better results.

## 5. Conclusions

The main purpose of the current study focused on a comprehensive evaluation of IMERG precipitation estimates in its three Early, Late and Final runs based on information from 186 automatic weather stations, managed by the Meteorological Service of Catalonia (NE Spain). The evaluation was carried out at different time scales (semi-hourly, hourly, daily, monthly, seasonal and annual) over a period of 6 years (2015–2020), based on the analysis of several metrics that quantify the error in precipitation accumulations. Similarly, the behaviour of IMERG was evaluated at a high resolution (30 min) under different topographic conditions (valley, flat, ridgetop), climatic conditions (BSk, Csa, Csb, Dfb) and under different precipitation intensity thresholds (light, moderate, heavy, very heavy, torrential). The main findings of the study are:

1.  IMERG generally captures the spatial–temporal pattern and variability of annual mean precipitation. However, discrepancies appear in the estimation of the magnitude. While IMERG_E and IMERG_L overestimate precipitation by 20% in practically the whole territory, IMERG_F reduces the error significantly, yielding only 2%. The calibration performance in this run may even cause an underestimation of precipitation in areas of complex orography such as the Pyrenees.
2.  The calculated statistics showed a significant improvement with decreasing temporal resolutions, with the monthly, seasonal and annual scales showing the best results in the estimation of precipitation accumulations. In contrast, the sub-daily scales showed high *Bias* values and very low correlation values, indicating the remaining challenge for satellite sensors to estimate precipitation at very high temporal resolutions. IMERG_F showed much better error statistics compared to IMERG_E and IMERG_L, wherein a generalised overestimation was evident and especially marked during the summer period.
3.  Similarly, the analysis of the *POD* and *FAR* showed a greater ability of IMERG to identify precipitation events at scales greater than daily, wherein a stable behaviour

of the statistics is observed well above the mean values, although with deficiencies in the identification of extreme events at all scales. The proportion of false alarms is a problem for IMERG especially during the summer, which is mainly associated with the detection of false precipitation in the form of lightrainfall (which is likely influenced by evaporation processes not assimilated by the algorithm), as well as the underestimation of locally occurring heavy precipitation.

4.  The worst results were obtained on a semi-hourly scale represented by flat areas and under a BSk climate, wherein IMERG shows a tendency to overestimate rainfall.

5.  IMERG tends to overestimate lightprecipitation, while it tends to underestimate accumulated precipitation in the rest of the intensity thresholds studied, especially those marked by high intensity precipitation. Associated with these errors is the fundamental role of taking rainfall gauges on a point scale that may not represent the spatial and temporal variability of rainfall in a region where this variable is spatially uncorrelated.

The evaluation of IMERG products presented here, although not the first one in Spain, is the first to address in detail the orographic and climatic factors at high temporal resolutions. Furthermore, we attempted to cover some of the most common weaknesses of this type of research by extending the analysis simultaneously to different temporal resolutions and by emphasising the analysis at high temporal resolutions. This study can be used by other researchers and developers involved in the IMERG algorithm to introduce improvements in future versions. Additionally, although with the limitation of latency, time observation and monitoring could be considered in operational work. For more applications based on the results presented here, and to try to answer some of the questions raised, in future work we intend to study in greater depth the capacity of IMERG to detect extreme events and to identify the specific behaviour of IMERG contributing sensors such as MW and IR.

**Author Contributions:** Conceptualization, E.P. and J.B.; methodology, E.P. and J.B.; formal analysis, E.P.; data curation, E.P.; writing—original draft preparation, E.P.; writing—review and editing, all the authors.; visualization, E.P.; supervision, J.B. and M.U. All authors have read and agreed to the published version of the manuscript.

**Funding:** This research was partly funded by the project "Analysis of Precipitation Processes in the Eastern Ebro Subbasin" (WISE-PreP, RTI2018-098693-B-C32 and ARTEMIS, PID2021-124253OB-I00 MINECO/FEDER) and the Water Research Institute (IdRA) of the University of Barcelona.

**Data Availability Statement:** IMERG data were downloaded from https://gpm.nasa.gov/data/directory and meteo.cat data were downloaded from https://www.meteo.cat/wpweb/serveis/cataleg-de-serveis/serveis-oberts/dades-obertes/.

**Acknowledgments:** Satellite and gauge data for this study were provided by NASA/JAXA and meteo.cat, respectively. The authors wish to thank the support from the Secretariat for Universities and Research of the Ministry of Business and Knowledge of the Government of Catalonia and the European Social Fund.

**Conflicts of Interest:** The authors declare no conflict of interest.

### Appendix A

Table A1 provides an overview of the data available for each temporal resolution considered in the study. The first column lists the maximum number of possible data records for each temporal resolution, calculated considering the number of existing stations for each year, which varies from 183 to 188 stations depending on the year. The second and third columns show the number and percentage of records verifying Criterion 1 (80% minimum availability of records needed for a given temporal period). The fourth and fifth column show the number and percentage of records verifying Criterion 2 (amounts equal to or higher than 0.1 mm for both rain gauge and IMERG products).

**Table A1.** Data availability for each temporal resolution considered in the study.

| Temporal Resolution | Maximum Number of Records | Criterion 1 | | Criterion 2 | |
|---|---|---|---|---|---|
| | | Number of Records | Percentage (%) | Number of Records | Percentage (%) |
| half-hourly | 19,482,432 | 18,804,667 | 97 | 277,616 | 1 |
| daily | 405,884 | 391,446 | 96 | 70,399 | 17 |
| monthly | 13,332 | 12,864 | 96 | 12,802 | 96 |
| spring | 1111 | 996 | 90 | 996 | 90 |
| summer | 1111 | 1020 | 92 | 1020 | 92 |
| autumn | 1111 | 1032 | 93 | 1032 | 93 |
| winter | 923 | 820 | 89 | 820 | 89 |
| annual | 1111 | 1034 | 93 | 1034 | 93 |

**Appendix B**

**Table A2.** Different climate areas of the Köppen climate classification [77–79] considered in this study.

| Code | Description | Group |
|---|---|---|
| BSk | Cold semi-arid (steppe) climate | Arid |
| Csa | Hot-summer Mediterranean climate | Temperate |
| Cf | Temperate without dry season | Temperate |
| Df | Continental without dry season | Cold (continental) |

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
