# Peer review of "Performance Assessment of GPM IMERG Products at Different Time Resolutions, Climatic Areas and Topographic Conditions in Catalonia"

_remotesensing, doi:10.3390/rs14205085_

Round 1
Reviewer 1 Report
The manuscript presented a comprehensive evaluation of IMERG (IMERG_E, _L, and _F) over the semiarid regions with complex orography over Catalonia. Since there are very few IMERG evaluation studies available at sub-daily scale, it would provide important information to the validation group and scientific community to identify the source of errors and thus to improve its performance in the upcoming versions. The manuscript is well written, and the methodological approach is good, considered different geographic conditions and different rainfall thresholds to reveal how the climatic and geographic typology affects the IMERG performances. The manuscript will be well suited to the scope of the journal, and readers of the journals will find it interesting. I would recommend it to consider for publication in the journal ‘Remote Sensing’ with a minor revision.
The followings are some comments and suggestions:
Line :73 “…compared GPM …” Please be specific weather it is IMERG or some other products from GPM as well.
Line :86 “…aiming to fill the research gap …” Even though the literatures were presented in the preceding paragraph, their limitations or the specific research gap you have identified and plan to address are missing. As it is the foundation for the entire study, I would recommend including a few lines here.
Line :94 What is the main reason for selecting this particular period of 2015-2020, given that IMERG data is available since early 2000?
Line :96 “subdaily” to “sub-daily”
Line :141 “infrared satellites” to “infrared sensor”.
Line :163 “…with a resolution of 0.1 mm…” it is not clear enough to which resolution is being referred to.
Line :196 “..considering only those records that marked at least 0.1 mm in both IMERG and XEMA..” Does this mean that this study only looked at rainy days (hit days) and excluded the non-rainy days from the analysis?
Line :197 Is it only referring to the IMERG-F final products? If so, please specify that.
Line 201 What is the source of elevation data (DEM) used for the study?
Line :217 “Fig :3 Distribution of IMERG stations” Is it IMERG pixels?
Line 268: Could you provide some references for the claim?
Line: 281 It is difficult to interpret what do you mean here, please make it clear. Also, are the authors referring to Figure.2 here (it doesn’t seem like support the text)?
Line: 286 Again, it seems like Figure.2 does not support the text here.
Line: 283 Were the authors' findings consistent with those of other studies that evaluated IMERG products in the same study area (IP)? Readers would be interested if the authors could compare their findings to those of other studies.
Line: 288 “..reduction of Bias by almost 90 %...” It is unclear whether the authors are referring to the actual bias (Rbias) or to the differences between the datasets in general. I'm not sure how to interpret the 90% bias reduction from Figure.4.
Line: 290-291 Is this implying that the IMERG E and L products underestimate precipitation in mountainous regions (-600 mm/year)? Until now, it appeared that both runs (_E and _L) overestimate precipitation while IMERG_F underestimates it. If I'm not interpreting the text correctly, please rewrite and elaborate those lines. It would also be easier for the readers if the author could be more specific about the isolated points where IMERG showed the greatest differences (it is a little bit hard to locate those points from Figure.4).
Line 442:444 Too long sentence and was not clear. It would be better if you rephrase it into two sentences.
Line 473: Light rainfall rates rainfall?
Line 512: “low resolution grid (1°x1°)”, Is it 0.1°x 0.1°?
Line 583: “..IMERG-F reduces the error significantly..” Given that the authors mentioned a 20% overestimation by IMERG-E and -L, it would be easy to follow if the authors could show numerically how much it was reduced for IMERG-F.

Reviewer 2 Report
In general the paper is good, but there are a few things I would like to clarify:
1. The key to ground validation of IMERG data is rain gauge observation. In this paper, the author must explain how the data quality process from the rain gauge has been carried out, because it is often found that the accuracy of the rain gauge itself is not good.
2. Another key to the ground validation of the IMERG data is the availability of observational data from each time scale tested. If the rain gauge observation is not 100%, then this is also a source of bias in ground validation. Therefore, the author should explain more about the availability of the data at each selected time scale.
3. Table 5 is not logical. The author uses 1 hour data for this table 5. If you look at table 4, CC for hourly is F (0.37), L (0.33), E (0.26). The values ​​in Table 5 are much smaller than Table 4 for all rainfall. There should be an intensity whose accuracy is better than the overall data in Table 4. Please check the correctness of the results in table 5.
Reviewer 3 Report
This manuscript presents a detailed evaluation of the IMERG products in Catalonia. It is a significant work. The manuscript is well written. Some minor comments are as follows.
1. Lines 16, 92: What do the abbreviations “BSk, Csa, Cf, Df” stand for? Please give a brief introduction.
2. Line 55: There is redundant “Dual Frequency”.
3. Line 78: “The” in “considered different the precipitation thresholds” should be removed.
4. Line 89: The semicolon makes the sentence confusing.
5. Line 154: Please give the full name for “IR”.
